# PMSPO: Progressive Matching and Semantic-Aware Policy Optimization for Camouflaged Object Detection

**Maosheng Su**[1]  **Junlei Cao**[1]  **Zhiyuan Wang**[1]  **Shuo Wang**[1]  **Ning Zhang**[1]  **Jun Luo**[1]

## Abstract

Reinforcement learning-based Multimodal Large Language Models (MLLMs) provide new perspectives for visual grounding, yet face significant challenges in Camouflaged Object Detection (COD) where objects blend seamlessly with backgrounds. This stems primarily from: difficulties in multi-object matching, the detrimental effects of low-quality samples, and erroneously localizing visual distractors with similar textures to true objects. We propose **P**rogressive **M**atching and **S**emantic-aware **P**olicy **O**ptimization (PMSPO), a curriculum learning-based framework that employs Sinkhorn multi-object matching IoU reward during training for multi-object alignment, utilizes Positive Learning Gain Filtering (PLGF) to curate high-quality samples, and transforms deep visual features into semantic contrastive reward rules to calibrate target background semantics. Experiments on COD benchmarks demonstrate that PMSPO achieves state-of-the-art (SOTA) performance among reinforcement learning methods across all evaluation metrics.

## 1. Introduction

Camouflaged object detection (COD) aims to identify objects that exhibit high similarity to their surrounding backgrounds in terms of texture, color, and shape (Fan et al., 2021a; Mei et al., 2021; Sun et al., 2021; Liang et al., 2024; Pang et al., 2024; Ji et al., 2023). This fundamental characteristic poses significant challenges for separating objects from complex backgrounds.

Recent advances in general Multimodal Large Language Models (MLLMs) have demonstrated remarkable zero-shot capabilities in visual understanding tasks. Models (Achiam et al., 2023; Dai et al., 2023; Liu et al., 2023; Bai et al., 2023; Chen et al., 2024; Deitke et al., 2025) can interpret complex visual scenes and generate localization cues through natural language interaction, offering potential for COD without task specific training. However, their perceptual performance degrades significantly in camouflaged scenarios, which model scaling alone cannot address. The fundamental reason lies in that these models are predominantly pretrained on datasets featuring "salient objects with clear foreground-background separation", resulting in visual priors that inherently conflict with camouflage characteristics. Consequently, when confronted with camouflaged scenes, these models tend to erroneously localize visual distractors that share similar textures or patterns with the background, as their learned representations favor high contrast, easily distinguishable objects rather than subtle, camouflaged objects. We term this problem semantic drift, where the model's predictions deviate from true camouflaged objects toward semantically confusing background regions.

To address this limitation, researchers have proposed prompt-based adaptation methods such as GenSAM (Hu et al., 2024a), ProMac (Hu et al., 2024b), VLCO (Su et al., 2025) and Lip (Zhang et al., 2025). These methods guide the model to localize camouflaged objects through carefully designed modules, without requiring task specific training on COD. Despite these efforts, the semantic drift problem persists: since these methods rely solely on the inherent capabilities of pretrained MLLMs without introducing additional discriminative mechanisms, they lack the ability to distinguish subtle differences between genuine objects and confusing background elements.

Beyond prompt-based methods, recent research has begun exploring the integration of MLLMs with reinforcement learning (Sutton et al., 1998; Mnih et al., 2015; Rafailov et al., 2023; Schulman et al., 2017) to address visual grounding tasks. Group Relative Policy Optimization (GRPO) (Shao et al., 2024; Guo et al., 2025) provides an effective framework that directly optimizes task-specific rewards without relying on explicit supervision signals. Methods such as VLM-R1 (Shen et al., 2025) and Seg-R1 (You & Wu, 2025) have demonstrated the potential of this paradigm by employing Intersection-over-Union (IoU) based reward

---

[1]College of Informatics, Huazhong Agricultural University, Wuhan, China. Correspondence to: Jun Luo <luojun@mail.hzau.edu.cn>.

*Proceedings of the 43rd International Conference on Machine Learning*, Seoul, South Korea. PMLR 306, 2026. Copyright 2026 by the author(s).

mechanisms, significantly improving localization accuracy. However, when these methods are applied to camouflaged object detection, critical problems emerge at multiple aspects. (i) For the multi-object matching problem, their matching strategies based on Hungarian algorithm (Kuhn, 1955; Carion et al., 2020) or greedy matching either produce hard assignments that block gradient propagation, or tend to fall into suboptimal solutions, limiting the effectiveness of reward-driven optimization in multi-instance scenarios. (ii) For the sample quality problem, these methods treat all training samples equally without considering the detrimental effects of low-quality samples, where noisy or ambiguous annotations can mislead the optimization process. (iii) For the semantic drift problem, they over-rely on simple geometric rewards, failing to capture semantic information and thus remaining susceptible to the semantic drift problem, causing the model to erroneously localize visual distractors with textures similar to true objects.

Based on the above analysis, we propose **P**rogressive **M**atching and **S**emantic-aware **P**olicy **O**ptimization (PM-SPO), a unified reinforcement learning framework that systematically address these three problems. For the matching problem, we introduce the Sinkhorn (Cuturi, 2013; Peyré & Cuturi, 2019; Tai et al., 2021) into MLLM-based multi-object matching for the first time, enabling stable gradient propagation through differentiable assignments. For the sample quality problem, we propose a Positive Learning Gain Filtering (PLGF) combined with a three-stage curriculum learning strategy, which filters detrimental samples and progressively increases task complexity to ensure stable policy optimization. For the semantic drift problem, we leverage the powerful semantic representation capability of DINOv2 (Oquab et al., 2023) to design a contrastive reward that effectively suppresses confusion with visual distractors by penalizing semantically inconsistent predictions. During inference, SAM3 (Carion et al., 2025) serves as the segmentation module to generate masks from predicted boxes.

In summary, we make the following contributions:

- We propose PMSPO, a reinforcement learning framework that combines PLGF with three-stage reward for camouflaged object detection.
- We propose a Sinkhorn multi-object matching IoU reward to address the multi-object matching problem, providing stable learning signals in multi-instance scenarios.
- To address semantic drift, we design a DINOv2-based semantic contrastive reward that penalizes predictions with high geometric overlap but semantic deviation from objects.
- Experiments on CHAMELEON (Skurowski et al., 2018), CAMO (Le et al., 2019), and COD10K (Fan et al., 2020) demonstrate that PMSPO achieves SOTA across all evaluation metrics.

## 2. Related Work

### 2.1. Visual Grounding with MLLMs

Visual grounding aims to localize objects in images based on natural language descriptions. Recent MLLMs (Achiam et al., 2023; Dai et al., 2023; Liu et al., 2023; Bai et al., 2023; Chen et al., 2024; Deitke et al., 2025) have achieved impressive performance on this task by directly generating bounding box coordinates in a text-to-text manner. These models benefit from large-scale vision-language pre-training that aligns visual regions with semantic concepts. However, their success largely depends on target objects exhibiting distinctive visual features separable from backgrounds, while camouflaged objects deliberately blend with their surroundings. When applied to such scenarios, these models tend to localize visually salient but semantically incorrect regions, exhibiting the semantic drift problem we identified.

### 2.2. MLLMs for Camouflaged Object Detection

To adapt MLLMs for camouflaged object detection, recent research has explored different strategies. GenSAM (Hu et al., 2024a) proposes Cross-modal Chains of Thought Prompting to decompose the detection process into reasoning steps, combined with Progressive Mask Generation that iteratively reweights image features to shift attention toward potential objects. ProMaC (Hu et al., 2024b) employs bidirectional cyclic iteration between prompts and masks, leveraging the reasoning capabilities of multimodal large models for continuous self-correction. VLCO (Su et al., 2025) designs dual-optimization prompts that fuse scene-level descriptions with fine-grained contour cues, enabling MLLMs to output pixel coordinates as segmentation prompts. LiP (Zhang et al., 2025) addresses the issue of subtle COD features by incorporating external knowledge. While these methods demonstrate the potential of MLLMs in camouflaged scenarios, they operate in a training-free and prompt-dependent paradigm, inherently limited by the pre-trained model's capability. Without learning from detection errors, these approaches remain susceptible to semantic drift when encountering visual distractors with similar textures.

### 2.3. Reinforcement Learning for Visual Localization

Reinforcement learning (Sutton et al., 1998; Mnih et al., 2015; Rafailov et al., 2023; Schulman et al., 2017) has emerged as a promising paradigm for enhancing visual understanding in MLLMs. GRPO (Shao et al., 2024) originally designed for mathematical reasoning, optimizes policies by comparing multiple sampled outputs within groups using rule-based rewards. This framework has been extended to visual tasks: VLM-R1 (Shen et al., 2025) applies GRPO with IoU-based rewards for object detection, while Seg-R1 (You & Wu, 2025) combines IoU with S-Measure rewards for

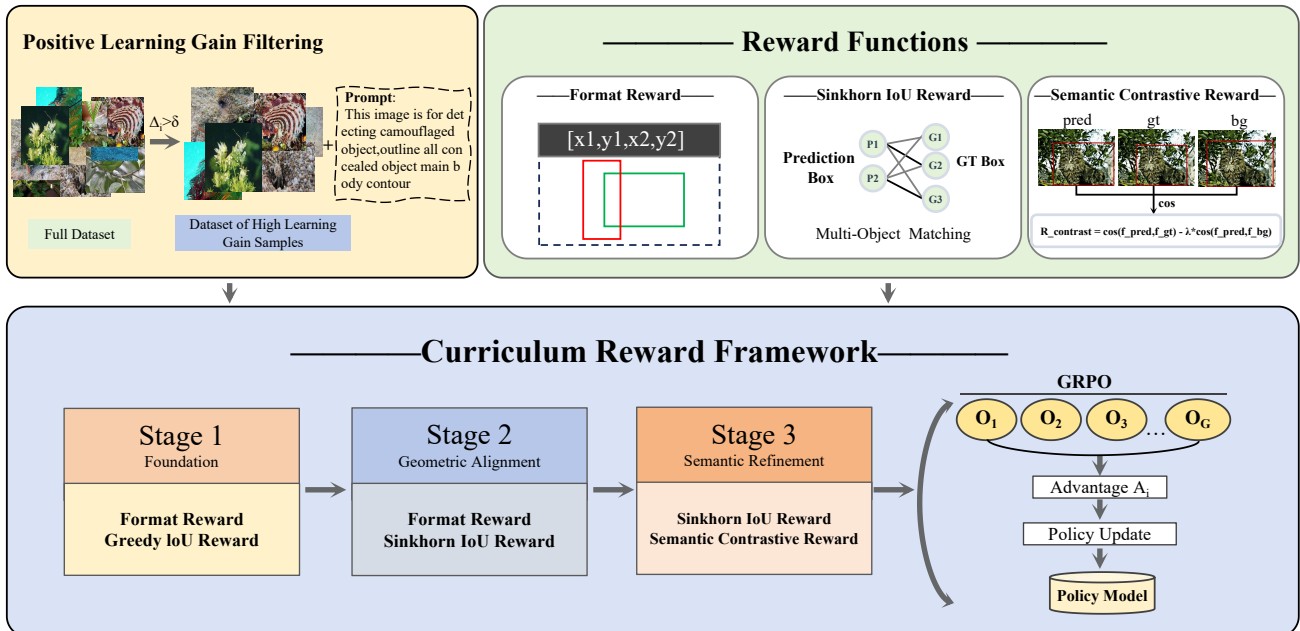

*Figure 1.* Our method comprises two main components. (i) **Positive Learning Gain Filtering** (top-left): We conduct one round of GRPO training with Sinkhorn IoU reward, then select high-quality samples based on reward improvement ($\Delta > \delta$). (ii) **Reward Functions** (top-right): Three complementary components: Format Reward for valid bounding box outputs, Sinkhorn IoU Reward for differentiable multi-object matching, and Semantic Contrastive Reward for background discrimination. These rewards are integrated through a **Curriculum Reward Framework** (bottom) across three progressive stages. The combined rewards drive GRPO-based policy optimization for enhanced camouflaged object detection.

camouflaged object segmentation. Visual-RFT (Liu et al., 2025) solves complex tasks by integrating explicit spatial localization into the full-cycle reasoning chain. However, these methods still face challenges in camouflaged scenarios: non-differentiable matching strategies limit multi-instance optimization, equal treatment of samples ignores annotation quality and sample difficulty variations, and purely geometric rewards fail to prevent semantic drift toward visually similar distractors. Our PMSPO framework addresses these problems through Sinkhorn differentiable matching, sample filtering, and semantic contrastive reward.

## 3. Method

**Preliminary.** Group Relative Policy Optimization (GRPO) (Shao et al., 2024) is a reinforcement learning algorithm that estimates advantages through group-wise comparisons, eliminating the need for a separate critic model. For each question $q$, GRPO samples a group of $G$ outputs $\{o_1, o_2, \dots, o_G\}$ from the old policy $\pi_{\theta_{old}}$, and computes the advantage for each output by normalizing the rewards within the group:

$$A_i = \frac{R(o_i) - \text{mean}(\{R(o_j)\}_{j=1}^{G})}{\text{std}(\{R(o_j)\}_{j=1}^{G})} \quad (1)$$

where $R(o_i)$ denotes the reward for output $i$. The policy is

then optimized by maximizing the following objective:

$$\mathcal{J}_{\text{GRPO}}(\theta) = \mathbb{E}_{q, \{o_i\}_{i=1}^{G}} \left[ \frac{1}{G} \sum_{i=1}^{G} \frac{1}{|o_i|} \sum_{t=1}^{|o_i|} \Big( \min\big(\rho_{i,t} A_i, \right.$$
$$\left. \text{clip}(\rho_{i,t}, 1-\epsilon, 1+\epsilon) A_i\big) - \beta D_{\text{KL}}(\pi_\theta \| \pi_{\text{ref}}) \Big) \right] \quad (2)$$

where $\rho_{i,t} = \frac{\pi_\theta(o_{i,t}|q, o_{i,<t})}{\pi_{\theta_{old}}(o_{i,t}|q, o_{i,<t})}$ denotes the importance sampling ratio, $\epsilon$ is the clipping parameter, $\beta$ is the KL regularization coefficient, and $\pi_{\text{ref}}$ is the reference policy.

**Overview.** The core of this work lies in designing the reward function $R(\cdot)$ in Eq. 1 for visual grounding in camouflaged scenarios. As illustrated in Fig. 1, our reward function comprises three complementary components: (i) Format Reward $R_{\text{format}}$ ensures outputs conform to valid bounding box format $[x_1, y_1, x_2, y_2]$; (ii) Sinkhorn multi-object matching IoU Reward $R_{\text{IoU}}$ enables differentiable multi-instance matching through assignment, providing smooth gradient signals for precise localization (Sec. 3.1); (iii) Semantic Contrastive Reward $R_{\text{contrast}}$ leverages visual features to prevent semantic drift (Sec. 3.3). These components are integrated through a Curriculum Reward Framework (Sec. 3.4) that progressively adjusts reward weights and combines multiple reward signals following a coarse-to-fine training paradigm,

ensuring stable optimization.

## 3.1. Sinkhorn Multi-Object Matching IoU Reward

In camouflaged object detection, an image may contain multiple target instances. The model predicts a set of bounding boxes $\mathcal{P} = \{\hat{\mathbf{b}}_1, \ldots, \hat{\mathbf{b}}_M\}$ while the ground truth is $\mathcal{G} = \{\mathbf{b}_1^{gt}, \ldots, \mathbf{b}_N^{gt}\}$. Before computing IoU rewards, we must first solve the assignment problem: determining which prediction corresponds to which ground truth.

**Limitations of Existing Strategies.** Greedy strategy assigns each prediction to its highest-IoU ground truth:

$$R_{\text{greedy}} = \frac{1}{M} \sum_{i=1}^{M} \max_{j \in [N]} \text{IoU}(\hat{b}_i, b_j^{gt}). \qquad (3)$$

However, this allows multiple predictions to match the same ground truth, rewarding redundant detections while failing to penalize missed objects. An alternative is the Hungarian algorithm, which enforces one-to-one matching but outputs binary assignments $\{0, 1\}$, creating a discontinuous reward landscape. In reinforcement learning, such discontinuity causes gradient issues: a nearly-correct prediction receives no learning signal if not selected.

To overcome limitations of existing strategies, we propose using the Sinkhorn algorithm to generate a soft assignment matrix. Given the pairwise IoU matrix $S \in \mathbb{R}^{M \times N}$ where $S_{ij} = \text{IoU}(\hat{b}_i, b_j^{gt})$, we first convert it to a cost matrix $C = \mathbf{1} - S$. To handle non-square cases where $M \neq N$, we pad the matrix to size $\max(M, N) \times \max(M, N)$ before applying Sinkhorn iterations. Let $P \in \mathbb{R}^{M \times N}$ denote the assignment matrix, where $P_{ij}$ represents the soft matching weight between prediction $i$ and ground truth $j$. The assignment matrix is initialized from the cost matrix and iteratively refined through alternating row and column normalization:

$$P^{(0)} = \exp(-C/\tau), \quad P^{(t+1)} = \mathcal{N}_r\big(\mathcal{N}_c(P^{(t)})\big), \quad (4)$$

where $P^{(0)}$ is the initial assignment derived from the exponentiated negative cost, $P^{(t)}$ denotes the assignment matrix at iteration $t$, $\tau$ is the temperature controlling assignment smoothness, and $\mathcal{N}_r(\cdot), \mathcal{N}_c(\cdot)$ denote row and column normalization respectively. After $T$ iterations, we truncate the result back to the original $M \times N$ dimensions to obtain the final assignment matrix $P^* \in [0, 1]^{M \times N}$. Unlike the binary assignments from Hungarian algorithm, $P^*$ provides soft weights that vary continuously with IoU values, enabling stable gradient propagation.

**Differentiable F-beta Reward.** With the assignment established, we define differentiable precision and recall. Precision $\widetilde{P}$ measures how much of each prediction's weight

falls on high-IoU ground truths:

$$\widetilde{P} = \frac{1}{M} \sum_{i=1}^{M} \sum_{j=1}^{N} P_{ij}^* \cdot S_{ij}. \qquad (5)$$

Recall $\widetilde{R}$ measures how well each ground truth is covered:

$$\widetilde{R} = \frac{1}{N} \sum_{j=1}^{N} \sum_{i=1}^{M} P_{ij}^* \cdot S_{ij}. \qquad (6)$$

We adopt the $F_\beta$ score as the final IoU reward function:

$$R_{\text{IoU}} = \frac{(1 + \beta^2) \cdot \widetilde{P} \cdot \widetilde{R}}{\beta^2 \cdot \widetilde{P} + \widetilde{R}}. \qquad (7)$$

Setting $\beta > 1$ emphasizes recall, which is crucial for camouflaged object detection where missing objects is typically more costly than false positives.

## 3.2. Positive Learning Gain Filtering

Although the Sinkhorn-based IoU reward provides smooth reward signals, not all training samples contribute positively to policy improvement. Certain samples, particularly those with highly ambiguous camouflage or annotation noise, yield negligible or negative reward changes after training, wasting computation and potentially introducing harmful gradients. If a sample's reward does not improve after training, it likely cannot provide useful learning signals. We propose PLGF as a data preprocessing step that measures the learning gain, the reward improvement achieved through one round of GRPO training with Sinkhorn-based reward, to identify and retain only beneficial samples.

**Filtering Procedure.** The filtering process consists of three steps. First, for each training sample $(\mathcal{I}_i, \mathcal{Q}_i)$, where $\mathcal{I}_i$ denotes the input image and $\mathcal{Q}_i$ denotes the text query, we predict bounding boxes using the base model $M_{\theta_0}$ and compute the Sinkhorn-based IoU reward:

$$R_i^{\text{before}} = R_{\text{IoU}}\big(M_{\theta_0}(\mathcal{I}_i, \mathcal{Q}_i), \mathcal{G}_i\big). \qquad (8)$$

Second, after one round of Sinkhorn-reward-based GRPO training on the complete dataset, we obtain model $M_{\theta_1}$ and compute updated rewards:

$$R_i^{\text{after}} = R_{\text{IoU}}\big(M_{\theta_1}(\mathcal{I}_i, \mathcal{Q}_i), \mathcal{G}_i\big). \qquad (9)$$

Third, we compute the learning gain $\Delta_i = R_i^{\text{after}} - R_i^{\text{before}}$ and retain samples exceeding a threshold $\delta$:

$$\mathcal{D}_{\text{filtered}} = \big\{(\mathcal{I}_i, \mathcal{Q}_i, \mathcal{G}_i) \mid \Delta_i > \delta\big\}. \qquad (10)$$

Discarded samples typically fall into three categories: those where the model already performs well with limited room

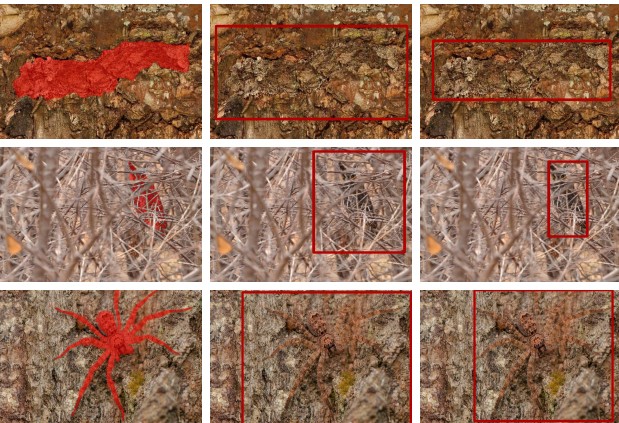

*Figure 2.* **Effectiveness of semantic contrastive reward.** Each row shows a camouflaged detection example. **Left**: Ground-truth mask (red). **Middle**: Prediction without contrast reward—boxes are overly large with excessive background. **Right**: Prediction with contrast reward—boxes tightly cover the target. This demonstrates that our reward guides the model to maximize target content while minimizing background inclusion.

for improvement; those with ambiguous annotations where the model cannot learn meaningful patterns; and those causing performance regression. The filtered dataset $\mathcal{D}_{\text{filtered}}$ is then used for all subsequent training stages.

### 3.3. Semantic Contrastive Reward

While the Sinkhorn-based IoU reward provides effective geometric supervision, it is inherently agnostic to semantic content. From a geometric perspective, a prediction that partially overlaps with a ground truth box appears reasonable and receives positive reward. However, as illustrated in Fig. 2, such a bounding box may capture wrong semantic content contaminated by background. This limitation is particularly critical in camouflaged scenarios, where the IoU metric cannot distinguish between a box correctly capturing a camouflaged animal and one capturing an adjacent salient background region with similar spatial extent.

**Feature Extraction.** We employ DINOv2 as the feature extractor due to its strong semantic discriminability. Given an input image $\mathcal{I}$ with spatial dimensions $H \times W$, we extract dense feature maps $F \in \mathbb{R}^{C \times H' \times W'}$ from the patch tokens, where $C$ is the feature channel dimension, and $H' = H/p, W' = W/p$ are the downsampled spatial dimensions with patch size $p$. For any bounding box $b$, we apply RoI Align (He et al., 2017) on $F$ followed by global average pooling to obtain a region-level feature vector $f_b \in \mathbb{R}^C$, which aggregates the spatial information within the box into a compact representation.

**Contrastive Formulation.** For each predicted box $\hat{b}$ matched to ground truth $b^{gt}$, we extract three feature vectors

from $F$:

$$f_{\text{pred}} = \phi(\hat{b}), \quad f_{\text{gt}} = \phi(b^{gt}), \quad f_{\text{bg}} = \phi(s \cdot b^{gt}), \quad (11)$$

where $\phi(\cdot)$ denotes the RoI feature extraction operation that maps a bounding box to a feature vector via RoI Align and pooling, and $s > 1$ is a scaling factor for the expanded region. Here, $f_{\text{pred}}$ represents what the model actually captures, $f_{\text{gt}}$ represents the pure target content, and $f_{\text{bg}}$ contains a mixture of target and surrounding background.

The semantic contrastive reward is defined as:

$$R_{\text{contrast}} = \cos(f_{\text{pred}}, f_{\text{gt}}) - \lambda \cdot \cos(f_{\text{pred}}, f_{\text{bg}}), \quad (12)$$

where $\cos(\cdot, \cdot)$ denotes the cosine similarity between two feature vectors, and $\lambda$ is a hyperparameter controlling the background penalty strength.

This formulation provides complementary supervision beyond geometric IoU. For correct predictions, $f_{\text{pred}}$ aligns closely with $f_{\text{gt}}$ while differing significantly from $f_{\text{bg}}$, yielding high contrastive reward. For drifted predictions, $f_{\text{pred}}$ deviates from $f_{\text{gt}}$ while becoming more similar to $f_{\text{bg}}$, resulting in suppressed or negative reward. This method effectively address semantic drift by penalizing predictions that capture salient background regions rather than true camouflaged targets.

### 3.4. Curriculum Reward Framework

Directly combining all reward components from the beginning leads to unstable optimization, as the model struggles to simultaneously learn valid bounding box generation, precise localization, and semantic discrimination. We propose a Curriculum (Bengio et al., 2009; Graves et al., 2017; Soviany et al., 2022) Reward Framework that progressively introduces components following a coarse-to-fine paradigm based on the model's learning readiness. Notably, all three stages are trained on the filtered dataset $\mathcal{D}_{\text{filtered}}$ obtained from PLGF (Sec. 3.2).

**Stage 1: Foundation Building (0–15% of training).** In the initial stage, model predictions are often chaotic with many outputs failing to generate valid bounding boxes. Thus, the primary goal is establishing stable bounding box generation capability. We assign high weight to the format reward $R_{\text{format}}$ to prioritize parseable outputs. For geometric supervision, we employ greedy matching rather than Sinkhorn at this stage, providing more forgiving gradient signals by rewarding any prediction that overlaps with some ground truth. This ensures even poor predictions receive feedback for improvement, laying the foundation for subsequent refinement.

**Stage 2: Geometric Alignment (15–40% of training).** Once the model reliably generates valid bounding boxes,

*Table 1.* Comparison with methods on three camouflaged object detection benchmarks. ↑ indicates higher is better, ↓ indicates lower is better. The best results are in **bold**.

| Method | Venue | CHAMELEON | | | | CAMO | | | | COD10K | | | |
|---|---|---|---|---|---|---|---|---|---|---|---|---|---|
| | | $M\downarrow$ | $F_\beta\uparrow$ | $E_\phi\uparrow$ | $S_\alpha\uparrow$ | $M\downarrow$ | $F_\beta\uparrow$ | $E_\phi\uparrow$ | $S_\alpha\uparrow$ | $M\downarrow$ | $F_\beta\uparrow$ | $E_\phi\uparrow$ | $S_\alpha\uparrow$ |
| *General MLLMs Method* | | | | | | | | | | | | | |
| Molmo-7B (Deitke et al., 2025) | CVPR25 | 0.092 | 0.583 | 0.663 | 0.643 | 0.143 | 0.468 | 0.572 | 0.563 | 0.086 | 0.438 | 0.612 | 0.638 |
| Qwen2.5-VL-3B (Bai et al., 2025) | Arxiv25 | 0.142 | 0.756 | 0.817 | 0.768 | 0.226 | 0.594 | 0.677 | 0.625 | 0.160 | 0.643 | 0.762 | 0.712 |
| Qwen2.5-VL-7B (Bai et al., 2025) | Arxiv25 | 0.063 | 0.826 | 0.889 | 0.838 | 0.144 | 0.716 | 0.785 | 0.732 | 0.056 | 0.807 | 0.894 | 0.842 |
| InternVL3.5-4B (Wang et al., 2025) | Arxiv25 | 0.085 | 0.664 | 0.746 | 0.723 | 0.155 | 0.523 | 0.658 | 0.610 | 0.056 | 0.708 | 0.832 | 0.774 |
| *MLLMs for COD Method* | | | | | | | | | | | | | |
| GenSAM (Hu et al., 2024a) | AAAI24 | 0.090 | 0.680 | 0.807 | 0.764 | 0.113 | 0.659 | 0.775 | 0.719 | 0.067 | 0.681 | 0.838 | 0.775 |
| ProMaC (Hu et al., 2024b) | NeurIps24 | 0.044 | 0.790 | 0.899 | 0.833 | 0.090 | 0.725 | 0.846 | 0.767 | 0.042 | 0.716 | 0.876 | 0.805 |
| VLCO (Su et al., 2025) | ICME25 | 0.045 | 0.800 | 0.872 | 0.825 | 0.089 | 0.756 | 0.818 | 0.775 | 0.037 | 0.769 | 0.874 | 0.820 |
| Lip (Zhang et al., 2025) | MM25 | 0.047 | 0.820 | 0.913 | 0.846 | 0.080 | 0.784 | 0.869 | 0.800 | 0.034 | 0.800 | 0.907 | 0.849 |
| *Reinforcement Learning Method* | | | | | | | | | | | | | |
| VLM-R1 (Shen et al., 2025) | Arxiv25 | 0.114 | 0.772 | 0.824 | 0.781 | 0.128 | 0.678 | 0.726 | 0.701 | 0.061 | 0.773 | 0.863 | 0.823 |
| Visual-RFT (Liu et al., 2025) | ICCV25 | 0.056 | 0.747 | 0.851 | 0.801 | 0.118 | 0.714 | 0.818 | 0.744 | 0.058 | 0.660 | 0.834 | 0.758 |
| Seg-R1-7B (You & Wu, 2025) | Arxiv25 | 0.040 | 0.836 | 0.915 | 0.868 | 0.073 | 0.788 | 0.881 | 0.826 | 0.031 | 0.820 | 0.926 | 0.873 |
| **PMSPO** | Ours | **0.036** | **0.875** | **0.921** | **0.878** | **0.057** | **0.834** | **0.885** | **0.833** | **0.024** | **0.852** | **0.935** | **0.879** |

we shift focus to localization accuracy. We transition to Sinkhorn matching with differentiable F-beta reward (Sec. 3.1), enabling fine-grained bounding box adjustment through smooth gradient signals. The format reward weight is reduced as output validity is no longer a bottleneck. During this stage, IoU scores improve rapidly as the model learns to spatially align with ground truth targets.

**Stage 3: Semantic Refinement (40–100% of training).** In the final stage, the model can generate geometrically aligned bounding boxes but may still suffer from semantic ambiguity, capturing regions that are spatially close to targets but semantically inconsistent. To address this, we introduce two complementary mechanisms.

First, applying nonlinear IoU sharpening via power transformation $R_{\text{IoU}}^\gamma$ where $\gamma > 1$ compresses low scores while amplifying high ones, pushing the model from approximate overlap to precise alignment.

Second, multiplicative contrastive modulation uses the semantic contrastive reward as a quality gate:

$$R_{\text{final}} = R_{\text{IoU}}^\gamma \cdot \left[1 + \alpha \cdot (R_{\text{contrast}} - 0.5) \times 2\right], \quad (13)$$

where $(R_{\text{contrast}} - 0.5) \times 2$ maps the contrastive score to $[-1, 1]$, and $\alpha$ controls modulation strength. This multiplicative formulation ensures that high-IoU predictions with correct semantics receive amplified rewards, while those with wrong semantics receive attenuated rewards, effectively preventing reward hacking.

This three-stage curriculum realizes a coarse-to-fine learning process, building capability from basic output generation to spatial precision to semantic understanding. By decomposing the complex learning objective into manageable sub-goals, our framework addresses the core challenges

in camouflaged object detection. This progressive training strategy enables the model to first establish robust detection capability before tackling the subtle discrimination between camouflaged targets and confusing backgrounds, ultimately achieving superior performance on COD tasks.

## 4. Experiments

### 4.1. Experimental Setup

**Datasets.** We evaluate our method on three widely-used camouflaged object detection benchmarks: COD10K (Fan et al., 2020), CAMO (Le et al., 2019), and CHAMELEON (Skurowski et al., 2018). We train on the combined training sets of COD10K and CAMO, and evaluate on all three test sets.

**Evaluation Metrics.** We adopt four standard metrics for camouflaged object detection: Structure-measure ($S_\alpha$) (Fan et al., 2017) evaluates structural similarity between predictions and ground truth; F-measure ($F_\beta$) (Margolin et al., 2014) balances precision and recall with adaptive thresholding; Mean E-measure ($E_\phi$) (Fan et al., 2021b) assesses both pixel-level and image-level accuracy; Mean Absolute Error ($M$) measures the average pixel-wise difference. Higher values indicate better performance for $S_\alpha$, $F_\beta$, and $E_\phi$, while lower $M$ is preferred.

**Implementation Details.** Our method is built upon Qwen2.5-VL-3B-Instruct (Bai et al., 2025) as the policy network. We incorporate a frozen DINOv2-ViT-G/14-reg (Oquab et al., 2023) as the visual feature extractor to compute semantic contrast rewards in Stage 3. Use SAM3 (Carion et al., 2025) to segment the bounding boxes output by the fine-tuned model to obtain masks. Training

*Table 2.* Ablation study on matching strategies.

| Matching | CHAMELEON | | | | CAMO | | | | COD10K | | | |
|---|---|---|---|---|---|---|---|---|---|---|---|---|
| | $M \downarrow$ | $F_\beta \uparrow$ | $E_\phi \uparrow$ | $S_\alpha \uparrow$ | $M \downarrow$ | $F_\beta \uparrow$ | $E_\phi \uparrow$ | $S_\alpha \uparrow$ | $M \downarrow$ | $F_\beta \uparrow$ | $E_\phi \uparrow$ | $S_\alpha \uparrow$ |
| Greedy | 0.054 | 0.842 | 0.893 | 0.848 | 0.078 | 0.800 | 0.858 | 0.802 | 0.039 | 0.827 | 0.906 | 0.853 |
| Hungarian | 0.052 | 0.848 | 0.891 | 0.844 | 0.080 | 0.801 | 0.856 | 0.805 | 0.041 | 0.830 | 0.911 | 0.858 |
| Sinkhorn | **0.048** | **0.853** | **0.906** | **0.857** | **0.073** | **0.809** | **0.863** | **0.811** | **0.035** | **0.839** | **0.918** | **0.861** |

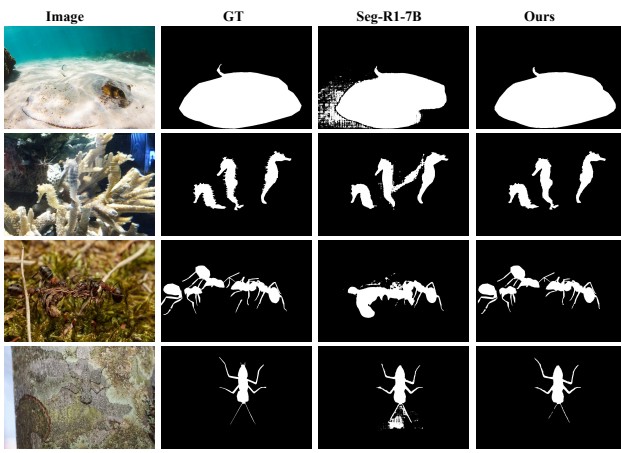

*Figure 3.* **Qualitative comparison on camouflaged object detection.** From left to right: input image, ground truth (GT), Seg-R1-7B prediction, and our prediction. Seg-R1-7B suffers from background interference leading to poor segmentation, while our method achieves more accurate boundary delineation and effectively suppresses false detections.

is conducted on 2 NVIDIA RTX PRO 6000 GPUs for a total of 1 epoch. Under the GRPO optimization framework, each prompt generates $G = 4$ candidate predictions for advantage estimation. Regarding reward function configuration, the Sinkhorn algorithm employs temperature $\tau = 0.1$ with $T = 20$ iterations to ensure stable convergence of the cost matrix toward a doubly stochastic matrix. To achieve better coverage in camouflaged scenarios, the F-beta reward is set to $\beta = 1.1$ to moderately increase the weight on recall. In the semantic refinement phase (Stage 3), we apply a power-law transformation with sharpening exponent $\gamma = 1.5$; meanwhile, the contrast modulation weight $\alpha$ linearly increases from 0.1 to 0.25 as training progresses. For semantic negative sampling, we expand the ground-truth box by a factor of $s = 1.5$ to delineate the background region, with penalty coefficient $\lambda_{\text{bg}} = 0.5$.

### 4.2. Comparison with SOTA Methods

We compare PMSPO with three categories of methods: (1) General MLLMs applied in zero-shot manner; (2) COD-specific MLLMs that optimize model outputs without task-specific training; and (3) Reinforcement learning methods that fine-tune MLLMs with reward signals. As shown in

Table 1, we achieve consistent improvements across all metrics on three benchmarks.

**Comparison with General MLLMs.** General MLLMs exhibit limited performance in camouflaged scenarios due to their pre-training domain gap. On COD10K, Qwen2.5-VL-3B achieves MAE of 0.160 and $S_\alpha$ of 0.712, while scaling to 7B parameters yields substantial improvements, reducing MAE from 0.160 to 0.056 by 65.0% and improving $S_\alpha$ from 0.712 to 0.842. This indicates that larger models possess stronger inherent capabilities for challenging visual tasks. However, our PMSPO using the same 3B base model achieves MAE of 0.024 and $S_\alpha$ of 0.879, representing an 85.0% MAE reduction compared to the 3B baseline and even 57.1% improvement over the 7B model. Similar trends are observed on CAMO and CHAMELEON. These results demonstrate that our framework with well-designed rewards can effectively bridge the domain gap, achieving performance gains that far exceed those obtained by simply scaling model parameters.

**Comparison with MLLMs for COD.** Methods specifically designed for COD, including approaches Gen-SAM (Hu et al., 2024a), ProMaC (Hu et al., 2024b), VLCO (Su et al., 2025), and Lip (Zhang et al., 2025), outperform general MLLMs but still fall short of our approach. On COD10K, Lip achieves the best baseline performance with MAE of 0.034. Our PMSPO outperforms Lip by a notable margin, reducing MAE from 0.034 to 0.024 by 29.4%. This gap reveals fundamental limitations of training-free methods: they cannot reshape representations biased toward salient objects from pre-training. Our approach directly optimizes model parameters through task-specific rewards, enabling genuine adaptation for camouflaged scenarios.

**Comparison with Reinforcement Learning Methods.** VLM-R1 shows poor performance across all datasets, with MAE of 0.114 on CHAMELEON, 0.128 on CAMO, and 0.061 on COD10K. This reveals that naively applying GRPO with simple IoU rewards leads to semantic drift, where models exploit rewards by localizing texture-similar distractors rather than true camouflaged targets.

We provide qualitative comparisons in Fig. 3. Seg-R1 (You & Wu, 2025) is a reinforcement learning method specifi-

*Table 3.* Ablation study on PLG filtering threshold $\delta$.

| Threshold $\delta$ | CHAMELEON | | | | CAMO | | | | COD10K | | | |
|---|---|---|---|---|---|---|---|---|---|---|---|---|
| | $M \downarrow$ | $F_\beta \uparrow$ | $E_\phi \uparrow$ | $S_\alpha \uparrow$ | $M \downarrow$ | $F_\beta \uparrow$ | $E_\phi \uparrow$ | $S_\alpha \uparrow$ | $M \downarrow$ | $F_\beta \uparrow$ | $E_\phi \uparrow$ | $S_\alpha \uparrow$ |
| 0.0 (no filtering) | 0.048 | 0.853 | 0.906 | 0.857 | 0.073 | 0.809 | 0.863 | 0.811 | 0.035 | 0.839 | 0.918 | 0.861 |
| 0.03 | **0.044** | **0.862** | **0.916** | **0.868** | **0.069** | **0.816** | **0.871** | **0.819** | **0.029** | **0.845** | **0.928** | **0.863** |
| 0.05 | 0.052 | 0.850 | 0.899 | 0.853 | 0.072 | 0.809 | 0.863 | 0.810 | 0.031 | 0.839 | 0.919 | 0.859 |
| 0.1 | 0.049 | 0.853 | 0.903 | 0.861 | 0.072 | 0.800 | 0.863 | 0.806 | 0.032 | 0.840 | 0.918 | 0.858 |

*Table 4.* Ablation study on component combination.

| Configuration | CHAMELEON | | | | CAMO | | | | COD10K | | | |
|---|---|---|---|---|---|---|---|---|---|---|---|---|
| | $M \downarrow$ | $F_\beta \uparrow$ | $E_\phi \uparrow$ | $S_\alpha \uparrow$ | $M \downarrow$ | $F_\beta \uparrow$ | $E_\phi \uparrow$ | $S_\alpha \uparrow$ | $M \downarrow$ | $F_\beta \uparrow$ | $E_\phi \uparrow$ | $S_\alpha \uparrow$ |
| Sinkhorn | 0.048 | 0.853 | 0.906 | 0.857 | 0.073 | 0.809 | 0.863 | 0.811 | 0.035 | 0.839 | 0.918 | 0.861 |
| + PLGF | 0.044 | 0.862 | 0.916 | 0.868 | 0.069 | 0.816 | 0.871 | 0.819 | 0.029 | 0.845 | 0.928 | 0.863 |
| + Semantic Contrastive Reward | 0.042 | 0.862 | 0.917 | 0.871 | 0.066 | 0.821 | 0.877 | 0.821 | 0.026 | 0.848 | 0.928 | 0.867 |
| + Curriculum Reward Framework | **0.036** | **0.875** | **0.921** | **0.878** | **0.057** | **0.834** | **0.885** | **0.833** | **0.024** | **0.852** | **0.935** | **0.879** |

cally designed for camouflaged object detection. However, despite using a 7B model with more than twice our parameters, it is consistently outperformed by our 3B PMSPO, which achieves 22.6% MAE reduction on COD10K. Our advantages stem from four key innovations: Sinkhorn matching for differentiable assignments, PLGF for high-quality sample curation, semantic contrastive reward for semantic consistency verification, and curriculum learning for progressive reward introduction. These demonstrate that reward design matters far more than model scale.

## 4.3. Ablation Studies

**Effect of Matching Strategies.** Table 2 compares three matching strategies. Greedy matching leads to suboptimal assignments when multiple predictions compete for multiple target. Hungarian matching resolves this but its binary assignments create gradient discontinuities. Our Sinkhorn matching generates continuous assignments, enabling smooth gradient flow. Sinkhorn consistently outperforms both baselines with $F_\beta$ improvements of +1.1%, +0.9%, and +1.2% on CHAMELEON, CAMO, and COD10K respectively, validating that differentiable assignments provide clearer optimization signals.

**Effect of PLGF Threshold.** Table 3 investigates the effect of different filtering thresholds $\delta$. Without filtering ($\delta = 0$), ambiguous samples introduce noisy gradients that destabilize optimization. The optimal threshold $\delta = 0.03$ achieves the best balance between data quality and quantity, reducing MAE by 12.1% on COD10K. At this threshold, the training set is refined from 4,040 to 1,768 samples (56.2% filtered), effectively removing images with subtle camouflage or annotation noise. However, aggressive filtering ($\delta \geq 0.05$) discards challenging samples essential for robust learning, ultimately leading to degraded generalization.

**Effect of Component Combination.** Table 4 presents the cumulative contribution of each component. Adding PLGF to Sinkhorn matching reduces MAE by 17.1% on COD10K. Directly adding semantic contrastive reward without curriculum scheduling yields inconsistent improvements, as abrupt semantic supervision interferes with geometric learning. The full curriculum framework unlocks semantic reward's potential with MAE reductions of 7.7%, demonstrating that the effectiveness of semantic supervision depends critically on when it is introduced during training.

**Additional Hyperparameter Studies.** We provide comprehensive ablation studies on remaining hyperparameters in the appendix: F-beta parameter $\beta$ (Appendix B.1), background expansion factor $s$ (Appendix B.2), background penalty coefficient $\lambda$ (Appendix B.3), IoU sharpening exponent $\gamma$ (Appendix B.4), contrast modulation strength $\alpha$ (Appendix B.5), curriculum stage boundaries (Appendix B.6), training epochs (Appendix B.7), reproducibility across random seeds (Appendix B.8), and visual feature extractor (Appendix B.9). These experiments validate that our default settings achieve optimal performance across three datasets.

## 5. Conclusion

We presented PMSPO, a progressive reinforcement learning framework for camouflaged object detection addressing three critical limitations: multi-object matching difficulty, low-quality sample interference, and semantic drift. Our approach introduces Sinkhorn multi-object matching IoU reward for differentiable assignment, Positive Learning Gain Filtering for data curation, and semantic contrastive reward leveraging DINOv2 to prevent semantic drift. These components are unified through a three-stage curriculum strategy from format compliance through geometric alignment to semantic refinement.

## Acknowledgements

This research was supported by the Fundamental Research Funds for the Central Universities (Grant No. 2662025PY019).

## Impact Statement

This paper presents work whose goal is to advance the field of Machine Learning. There are many potential societal consequences of our work, none which we feel must be specifically highlighted here.

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

# A. Algorithm of PMSPO

We present the complete PMSPO training algorithm in Algorithm 1, which consists of two phases: Positive Learning Gain Filtering (PLGF) and Curriculum Reward Framework. In Phase 1, we employ the Sinkhorn Multi-Object Matching IoU Reward (Sec. 3.1) to evaluate sample quality and filter the training data. In Phase 2, the curriculum reward framework is trained exclusively on the filtered dataset $\mathcal{D}_{\text{filtered}}$ obtained from Phase 1.

---

**Algorithm 1** PMSPO: Progressive Matching and Semantic-aware Policy Optimization

---

**Require:** Dataset $\mathcal{D}$, base model $M_{\theta_0}$, threshold $\delta$, total steps $T_{\text{total}}$
**Ensure:** Trained policy $\pi_\theta$
1: **// Phase 1: Positive Learning Gain Filtering**
2: Compute $R_i^{\text{before}} = R_{\text{sinkhorn}}(M_{\theta_0}(\mathcal{I}_i, \mathcal{Q}_i), \mathcal{G}_i)$ for all samples
3: Train one GRPO round with $R_{\text{sinkhorn}}$ on $\mathcal{D} \rightarrow M_{\theta_1}$
4: Compute $R_i^{\text{after}} = R_{\text{sinkhorn}}(M_{\theta_1}(\mathcal{I}_i, \mathcal{Q}_i), \mathcal{G}_i)$ for all samples
5: $\mathcal{D}_{\text{filtered}} \leftarrow \{(\mathcal{I}_i, \mathcal{Q}_i, \mathcal{G}_i) \mid R_i^{\text{after}} - R_i^{\text{before}} > \delta\}$
6: **// Phase 2: Curriculum Reward Framework**
7: **for** step $t = 1$ to $T_{\text{total}}$ **do**
8:     $p \leftarrow t/T_{\text{total}}$ {Training progress}
9:     Sample batch $\mathcal{B}$ from $\mathcal{D}_{\text{filtered}}$; for each input, sample $G$ outputs $\{o^j\}$
10:     **for** each output $o^j$ **do**
11:         **if** $p < 0.15$ **then**
            {Stage 1: Foundation Building}
12:             $s \leftarrow p/0.15$ {Stage progress}
13:             $w_f \leftarrow 0.5 \cdot (1 - 0.5s), \quad w_{\text{iou}} \leftarrow 0.5 + 0.25s$ {$w_f$: $0.5 \rightarrow 0.25$, $w_{\text{iou}}$: $0.5 \rightarrow 0.75$}
14:             $R^j \leftarrow w_f \cdot R_{\text{fmt}}^j + w_{\text{iou}} \cdot R_{\text{greedy}}^j$
15:         **else if** $p < 0.40$ **then**
            {Stage 2: Geometric Alignment}
16:             $s \leftarrow (p - 0.15)/0.25$
17:             $w_f \leftarrow 0.25 \cdot (1 - s), \quad w_{\text{iou}} \leftarrow 0.75 + 0.25s$ {$w_f$: $0.25 \rightarrow 0$, $w_{\text{iou}}$: $0.75 \rightarrow 1.0$}
18:             $R^j \leftarrow w_f \cdot R_{\text{fmt}}^j + w_{\text{iou}} \cdot R_{\text{sinkhorn}}^j$
19:         **else**
            {Stage 3: Semantic Refinement}
20:             $s \leftarrow (p - 0.40)/0.60$
21:             $\alpha \leftarrow 0.1 + 0.15s$ {Contrast weight: $0.1 \rightarrow 0.25$}
22:             $R^j \leftarrow (R_{\text{sinkhorn}}^j)^\gamma \cdot [1 + \alpha \cdot (R_{\text{contrast}}^j - 0.5) \times 2]$
23:         **end if**
24:     **end for**
25:     Compute advantages $A^j$ and update via GRPO
26: **end for**$\pi_\theta$

---

Here, $R_{\text{sinkhorn}}$ denotes the Sinkhorn Multi-Object Matching IoU Reward (Sec. 3.1), which is used in both Phase 1 for data filtering and Phase 2 for training. $R_{\text{contrast}}$ denotes the semantic contrastive reward (Sec. 3.3). Note that Phase 2 operates exclusively on the filtered dataset $\mathcal{D}_{\text{filtered}}$, ensuring that only high-quality samples contribute to the curriculum training.

# B. Additional Ablation Studies

## B.1. Effect of F-beta Parameter $\beta$

**Corresponds to:** Section 3.1 *Sinkhorn Multi-Object Matching IoU Reward*, Eq. (7). The parameter $\beta$ in the $F_\beta$ score controls the trade-off between precision and recall in the differentiable matching reward.

**Analysis.** As shown in Table 5, the optimal $\beta = 1.1$ slightly favors recall over precision, achieving the best performance across all datasets. This recall bias is critical for COD where missing targets is more costly than false positives. Precision-focused settings ($\beta = 0.8$) yield higher MAE (0.040 on CHAMELEON, 0.063 on CAMO) due to conservative predictions

*Table 5.* Ablation study on $\beta$ value in F-beta reward.

| $\beta$ | CHAMELEON | | | | CAMO | | | | COD10K | | | |
|---|---|---|---|---|---|---|---|---|---|---|---|---|
| | $M \downarrow$ | $F_\beta \uparrow$ | $E_\phi \uparrow$ | $S_\alpha \uparrow$ | $M \downarrow$ | $F_\beta \uparrow$ | $E_\phi \uparrow$ | $S_\alpha \uparrow$ | $M \downarrow$ | $F_\beta \uparrow$ | $E_\phi \uparrow$ | $S_\alpha \uparrow$ |
| 0.8 | 0.040 | 0.865 | 0.912 | 0.868 | 0.063 | 0.822 | 0.874 | 0.822 | 0.027 | 0.846 | 0.928 | 0.871 |
| 1.0 | 0.038 | 0.870 | 0.916 | 0.873 | 0.060 | 0.828 | 0.880 | 0.828 | 0.026 | 0.849 | 0.931 | 0.875 |
| **1.1** | **0.036** | **0.875** | **0.921** | **0.878** | **0.057** | **0.834** | **0.885** | **0.833** | **0.024** | **0.852** | **0.935** | **0.879** |
| 1.5 | 0.040 | 0.867 | 0.913 | 0.869 | 0.062 | 0.824 | 0.876 | 0.824 | 0.027 | 0.845 | 0.927 | 0.870 |

that miss target parts, while over-emphasizing recall ($\beta = 1.5$) also degrades performance (MAE increases to 0.040, 0.062, 0.027) as it produces oversized boxes with excessive background.

## B.2. Effect of Background Expansion Factor $s$

**Corresponds to:** Section 3.3 *Semantic Contrastive Reward*, Eq. (11). The expansion factor $s$ determines the size of the background region used for contrastive feature extraction.

*Table 6.* Ablation study on background expansion factor $s$.

| $s$ | CHAMELEON | | | | CAMO | | | | COD10K | | | |
|---|---|---|---|---|---|---|---|---|---|---|---|---|
| | $M \downarrow$ | $F_\beta \uparrow$ | $E_\phi \uparrow$ | $S_\alpha \uparrow$ | $M \downarrow$ | $F_\beta \uparrow$ | $E_\phi \uparrow$ | $S_\alpha \uparrow$ | $M \downarrow$ | $F_\beta \uparrow$ | $E_\phi \uparrow$ | $S_\alpha \uparrow$ |
| 1.2 | 0.041 | 0.866 | 0.912 | 0.867 | 0.064 | 0.821 | 0.873 | 0.819 | 0.027 | 0.846 | 0.928 | 0.871 |
| **1.5** | **0.036** | **0.875** | **0.921** | **0.878** | **0.057** | **0.834** | **0.885** | **0.833** | **0.024** | **0.852** | **0.935** | **0.879** |
| 1.8 | 0.040 | 0.867 | 0.913 | 0.869 | 0.062 | 0.825 | 0.876 | 0.823 | 0.027 | 0.847 | 0.929 | 0.872 |

**Analysis.** As shown in Table 6, the expansion factor $s = 1.5$ achieves the best performance, capturing sufficient local context for effective foreground-background discrimination. When $s = 1.2$, the expanded region is too similar to the target, reducing discriminative power (MAE increases to 0.041, 0.064, 0.027). When $s = 1.8$, the expanded region introduces semantically unrelated distant background, adding noise to contrastive learning (MAE of 0.040, 0.062, 0.027).

## B.3. Effect of Background Penalty Coefficient $\lambda$

**Corresponds to:** Section 3.3 *Semantic Contrastive Reward*, Eq. (12). The coefficient $\lambda$ controls the strength of background penalty in the contrastive reward formulation.

*Table 7.* Ablation study on background penalty coefficient $\lambda$.

| $\lambda$ | CHAMELEON | | | | CAMO | | | | COD10K | | | |
|---|---|---|---|---|---|---|---|---|---|---|---|---|
| | $M \downarrow$ | $F_\beta \uparrow$ | $E_\phi \uparrow$ | $S_\alpha \uparrow$ | $M \downarrow$ | $F_\beta \uparrow$ | $E_\phi \uparrow$ | $S_\alpha \uparrow$ | $M \downarrow$ | $F_\beta \uparrow$ | $E_\phi \uparrow$ | $S_\alpha \uparrow$ |
| 0.3 | 0.039 | 0.868 | 0.914 | 0.871 | 0.062 | 0.826 | 0.878 | 0.825 | 0.027 | 0.847 | 0.930 | 0.873 |
| **0.5** | **0.036** | **0.875** | **0.921** | **0.878** | **0.057** | **0.834** | **0.885** | **0.833** | **0.024** | **0.852** | **0.935** | **0.879** |
| 0.7 | 0.038 | 0.871 | 0.917 | 0.874 | 0.059 | 0.830 | 0.881 | 0.829 | 0.025 | 0.849 | 0.932 | 0.876 |
| 1.0 | 0.041 | 0.865 | 0.912 | 0.868 | 0.063 | 0.823 | 0.875 | 0.821 | 0.028 | 0.844 | 0.927 | 0.871 |

**Analysis.** As shown in Table 7, the optimal $\lambda = 0.5$ balances target inclusion and background exclusion. With weak penalty ($\lambda = 0.3$), predictions tend to include more background (MAE increases to 0.039, 0.062, 0.027). With aggressive penalty ($\lambda = 1.0$), boxes become overly tight and miss target parts, leading to the worst performance (MAE of 0.041, 0.063, 0.028), particularly degrading on CHAMELEON where targets have irregular boundaries.

## B.4. Effect of IoU Sharpening Exponent $\gamma$

**Corresponds to:** Section 3.4 *Curriculum Reward Framework*, Stage 3: Semantic Refinement, Eq. (13). The exponent $\gamma$ applies nonlinear sharpening to IoU rewards to encourage precise predictions.

*Table 8.* Ablation study on IoU sharpening exponent $\gamma$.

| $\gamma$ | CHAMELEON | | | | CAMO | | | | COD10K | | | |
|---|---|---|---|---|---|---|---|---|---|---|---|---|
| | $M \downarrow$ | $F_\beta \uparrow$ | $E_\phi \uparrow$ | $S_\alpha \uparrow$ | $M \downarrow$ | $F_\beta \uparrow$ | $E_\phi \uparrow$ | $S_\alpha \uparrow$ | $M \downarrow$ | $F_\beta \uparrow$ | $E_\phi \uparrow$ | $S_\alpha \uparrow$ |
| 1.0 | 0.041 | 0.866 | 0.913 | 0.867 | 0.063 | 0.823 | 0.875 | 0.821 | 0.027 | 0.847 | 0.929 | 0.872 |
| 1.2 | 0.039 | 0.870 | 0.916 | 0.872 | 0.061 | 0.827 | 0.879 | 0.826 | 0.026 | 0.849 | 0.932 | 0.875 |
| **1.5** | **0.036** | **0.875** | **0.921** | **0.878** | **0.057** | **0.834** | **0.885** | **0.833** | **0.024** | **0.852** | **0.935** | **0.879** |
| 2.0 | 0.037 | 0.872 | 0.918 | 0.875 | 0.059 | 0.831 | 0.882 | 0.830 | 0.025 | 0.850 | 0.933 | 0.876 |

**Analysis.** As shown in Table 8, moderate sharpening ($\gamma = 1.5$) creates a nonlinear reward landscape that strongly incentivizes precise predictions. Both insufficient ($\gamma = 1.0$) and excessive ($\gamma = 2.0$) sharpening degrade performance via coarse boundaries and optimization instability, respectively.

### B.5. Effect of Contrast Modulation Strength $\alpha$

**Corresponds to:** Section 3.4 *Curriculum Reward Framework*, Stage 3: Semantic Refinement, Eq. (13). The parameter $\alpha$ controls the multiplicative modulation strength of semantic contrastive reward.

*Table 9.* Ablation study on contrast modulation strength $\alpha$.

| $\alpha$ | CHAMELEON | | | | CAMO | | | | COD10K | | | |
|---|---|---|---|---|---|---|---|---|---|---|---|---|
| | $M \downarrow$ | $F_\beta \uparrow$ | $E_\phi \uparrow$ | $S_\alpha \uparrow$ | $M \downarrow$ | $F_\beta \uparrow$ | $E_\phi \uparrow$ | $S_\alpha \uparrow$ | $M \downarrow$ | $F_\beta \uparrow$ | $E_\phi \uparrow$ | $S_\alpha \uparrow$ |
| 0.0 | 0.044 | 0.862 | 0.911 | 0.858 | 0.066 | 0.817 | 0.868 | 0.814 | 0.029 | 0.845 | 0.928 | 0.863 |
| 0.1 (fixed) | 0.041 | 0.867 | 0.915 | 0.866 | 0.063 | 0.823 | 0.874 | 0.821 | 0.027 | 0.848 | 0.931 | 0.871 |
| 0.2 (fixed) | 0.039 | 0.870 | 0.917 | 0.871 | 0.061 | 0.828 | 0.879 | 0.826 | 0.026 | 0.850 | 0.933 | 0.875 |
| 0.3 (fixed) | 0.040 | 0.868 | 0.914 | 0.868 | 0.062 | 0.825 | 0.876 | 0.823 | 0.027 | 0.847 | 0.929 | 0.872 |
| **0.1→0.25** | **0.036** | **0.875** | **0.921** | **0.878** | **0.057** | **0.834** | **0.885** | **0.833** | **0.024** | **0.852** | **0.935** | **0.879** |

**Analysis.** As shown in Table 9, the linear schedule (0.1→0.25) significantly outperforms all fixed values. Compared to no contrast ($\alpha = 0.0$), the progressive schedule reduces MAE by 18.2%, 13.6%, and 17.2% on three datasets respectively. This progressive strategy allows the model to first establish geometric competence before emphasizing semantic discrimination, avoiding early-stage instability from premature semantic guidance.

### B.6. Effect of Curriculum Stage Boundaries

**Corresponds to:** Section 3.4 *Curriculum Reward Framework*, covering all three stages: Stage 1 (Foundation Building, 0–15%), Stage 2 (Geometric Alignment, 15–40%), and Stage 3 (Semantic Refinement, 40–100%).

*Table 10.* Ablation study on curriculum stage boundaries.

| S1→S2 | S2→S3 | CHAMELEON | | | | CAMO | | | | COD10K | | | |
|---|---|---|---|---|---|---|---|---|---|---|---|---|---|
| | | $M \downarrow$ | $F_\beta \uparrow$ | $E_\phi \uparrow$ | $S_\alpha \uparrow$ | $M \downarrow$ | $F_\beta \uparrow$ | $E_\phi \uparrow$ | $S_\alpha \uparrow$ | $M \downarrow$ | $F_\beta \uparrow$ | $E_\phi \uparrow$ | $S_\alpha \uparrow$ |
| 5% | 25% | 0.046 | 0.854 | 0.901 | 0.852 | 0.070 | 0.810 | 0.862 | 0.808 | 0.031 | 0.838 | 0.919 | 0.859 |
| 10% | 30% | 0.041 | 0.866 | 0.913 | 0.867 | 0.063 | 0.824 | 0.875 | 0.822 | 0.027 | 0.847 | 0.929 | 0.872 |
| **15%** | **40%** | **0.036** | **0.875** | **0.921** | **0.878** | **0.057** | **0.834** | **0.885** | **0.833** | **0.024** | **0.852** | **0.935** | **0.879** |
| 25% | 60% | 0.038 | 0.871 | 0.917 | 0.874 | 0.059 | 0.830 | 0.881 | 0.828 | 0.025 | 0.850 | 0.932 | 0.876 |
| 35% | 70% | 0.041 | 0.866 | 0.912 | 0.868 | 0.063 | 0.823 | 0.874 | 0.821 | 0.027 | 0.846 | 0.928 | 0.871 |

**Analysis.** As shown in Table 10, the optimal boundaries (15%→40%) allocate 15% for format learning, 25% for geometric alignment, and 60% for semantic refinement. Early transitions (5%→25%) cause the worst performance due to insufficient foundation building, leading to unstable optimization. Late transitions (35%→70%) also degrade performance as they reduce time for semantic refinement, limiting the final performance.

## B.7. Effect of Training Epochs

Training Configuration. We investigate whether training for additional epochs yields further performance gains, as GRPO-based reinforcement learning may in principle benefit from extended optimization.

*Table 11.* Ablation study on number of training epochs.

| Epochs | CHAMELEON | | | | CAMO | | | | COD10K | | | |
|---|---|---|---|---|---|---|---|---|---|---|---|---|
| | $M \downarrow$ | $F_\beta \uparrow$ | $E_\phi \uparrow$ | $S_\alpha \uparrow$ | $M \downarrow$ | $F_\beta \uparrow$ | $E_\phi \uparrow$ | $S_\alpha \uparrow$ | $M \downarrow$ | $F_\beta \uparrow$ | $E_\phi \uparrow$ | $S_\alpha \uparrow$ |
| **1** | **0.036** | **0.875** | **0.921** | **0.878** | **0.057** | **0.834** | **0.885** | **0.833** | **0.024** | **0.852** | **0.935** | **0.879** |
| 2 | 0.041 | 0.871 | 0.920 | 0.866 | 0.065 | 0.821 | 0.874 | 0.824 | 0.028 | 0.846 | 0.932 | 0.877 |

**Analysis.** As shown in Table 11, training for 2 epochs consistently degrades performance across all three datasets while doubling the training cost. Specifically, MAE increases by 13.9%, 14.0%, and 16.7% on CHAMELEON, CAMO, and COD10K, respectively. Our single-epoch curriculum is therefore both computationally efficient and empirically optimal.

## B.8. Reproducibility Across Random Seeds

Training Configuration. We assess the sensitivity of PMSPO to random initialization by conducting 4 independent runs with different random seeds across all three benchmark datasets.

*Table 12.* Ablation study on independent runs with different random seeds.

| Seed | CHAMELEON | | | | CAMO | | | | COD10K | | | |
|---|---|---|---|---|---|---|---|---|---|---|---|---|
| | $M \downarrow$ | $F_\beta \uparrow$ | $E_\phi \uparrow$ | $S_\alpha \uparrow$ | $M \downarrow$ | $F_\beta \uparrow$ | $E_\phi \uparrow$ | $S_\alpha \uparrow$ | $M \downarrow$ | $F_\beta \uparrow$ | $E_\phi \uparrow$ | $S_\alpha \uparrow$ |
| 42 (ours) | 0.036 | 0.875 | 0.921 | 0.878 | 0.057 | 0.834 | 0.885 | 0.833 | 0.024 | 0.852 | 0.935 | 0.879 |
| 123 | 0.036 | 0.876 | 0.922 | 0.879 | 0.057 | 0.836 | 0.887 | 0.831 | 0.024 | 0.853 | 0.936 | 0.880 |
| 566 | 0.037 | 0.874 | 0.920 | 0.877 | 0.056 | 0.831 | 0.882 | 0.833 | 0.025 | 0.850 | 0.933 | 0.877 |
| 2026 | 0.036 | 0.875 | 0.921 | 0.878 | 0.059 | 0.833 | 0.881 | 0.834 | 0.024 | 0.852 | 0.935 | 0.879 |

**Analysis.** As shown in Table 12, all standard deviations remain $\leq 0.003$ across all metrics and datasets, demonstrating that PMSPO is robust to random initialization. The negligible variance on CHAMELEON, CAMO, and COD10K confirms that the observed improvements stem from our architectural and reward design rather than favorable random seeds. This stability is attributable to the curriculum reward framework, which imposes structured learning stages that reduce sensitivity to initialization conditions.

## B.9. Effect of Visual Feature Extractor

The semantic contrastive reward relies on a frozen visual backbone to extract foreground and background features. We examine whether upgrading to a stronger extractor yields performance gains, validating the generality of our framework.

*Table 13.* Ablation study on visual feature extractor.

| Extractor | CHAMELEON | | | | CAMO | | | | COD10K | | | |
|---|---|---|---|---|---|---|---|---|---|---|---|---|
| | $M \downarrow$ | $F_\beta \uparrow$ | $E_\phi \uparrow$ | $S_\alpha \uparrow$ | $M \downarrow$ | $F_\beta \uparrow$ | $E_\phi \uparrow$ | $S_\alpha \uparrow$ | $M \downarrow$ | $F_\beta \uparrow$ | $E_\phi \uparrow$ | $S_\alpha \uparrow$ |
| DINOv2-ViT-G/14-reg | 0.036 | 0.875 | 0.921 | 0.878 | 0.057 | 0.834 | 0.885 | 0.833 | 0.024 | 0.852 | 0.935 | 0.879 |
| DINOv3-ViT-H$^+$/16 (distilled) | 0.033 | 0.882 | 0.928 | 0.885 | 0.053 | 0.842 | 0.892 | 0.841 | 0.021 | 0.860 | 0.941 | 0.886 |

**Analysis.** As shown in Table 13, replacing DINOv2-ViT-G/14-reg with the stronger DINOv3-ViT-H$^+$/16 (distilled) backbone yields consistent improvements across all datasets, reducing MAE by 8.3%, 7.0%, and 12.5% on CHAMELEON, CAMO, and COD10K, respectively. This confirms the generality of our framework: a stronger visual extractor naturally provides higher-quality reward signals for foreground-background discrimination, which is precisely the role the semantic contrastive reward is designed to exploit. The framework is thus backbone-agnostic and scales gracefully with extractor capacity.

