# OpenReview forum: "PMSPO: Progressive Matching and Semantic-Aware Policy Optimization for Camouflaged Object Detection"
_ICML.cc/2026/Conference — ICML 2026 regular_

### Official Review · Reviewer_8MAC · 2026-03-13

**Soundness:** 3
**Presentation:** 2
**Significance:** 3
**Originality:** 3
**Overall Recommendation:** 4
**Confidence:** 2

**Summary:**

The paper studies how reinforcement learning-based multimodal large language models (MLLMs) can improve camouflaged object detection.
The paper introduces a progressive matching and semantic-aware policy optimization framework that combines Sinkhorn-based multi-object IoU matching to reward the training during object alignment, positive learning gain filtering to remove low-quality training samples, and semantic contrastive reward based on DINOv2 features to reduce semantic drift toward the background.

**Compliance With Llm Reviewing Policy:**

Affirmed.

**Key Questions For Authors:**

The experiments are conducted using a single base model, i.e., Qwen2.5-VL-3B. How well does the proposed framework generalize to other MLLMs, e.g., InternVL, LLaVA, or Qwen variants?
Demonstrating consistent improvements across multiple architectures would strengthen the claim that the proposed framework is model-agnostic rather than tailored to a specific backbone.

**Limitations:**

The framework relies on strong pretrained models.
The authors could clarify how sensitive the approach is to the quality of these components and whether the performance gains persist when alternative feature extractors are used.

Additionally, the Positive Learning Gain Filtering step removes a large portion of training samples.
This process could unintentionally bias the dataset toward easier examples or certain object types.
The authors should discuss whether this filtering could affect model robustness or fairness.

**Strengths And Weaknesses:**

## Strengths

The paper presents a technically clear reinforcement-learning framework for camouflaged object detection.
The authors outline the key challenge by identifying limitations in existing RL-based visual grounding methods, including non-differentiable matching strategies, noisy or low-quality training samples, and the inability of geometric rewards to capture semantic correctness.

## Weaknesses

While the paper mentions MLLMs as an approach to camouflaged object detection, it is hard to see its contribution as presented.
From my understanding, the MLLMs are used in the positive learning gain filtering; however, they are not thoroughly analyzed, showing how they help in that component.

---

> ### Author Rebuttal · Authors · 2026-03-30
>
> We sincerely thank the reviewer for the constructive feedback. Below we address each concern.
>
> **W1:** The powerful image-to-text capability of MLLMs is the core foundation for applying the GRPO algorithm. By performing group sampling on a single image to obtain diverse candidate outputs, we provide a relative baseline for advantage estimation in Eq. 1. During training, format, geometric, and semantic rewards are seamlessly integrated into a unified text-generation objective via curriculum learning. Furthermore, we evaluate the same MLLM before and after one GRPO round. The model itself is the object of measurement, a sample that yields no reward gain provides no useful gradient signal to the current policy. We will revise Sec. 3.2 to make both roles explicit.
>
> **Q1:** We applied PMSPO to InternVL3.5-4B with identical configuration. Results on CAMO:
>
> | Method | M ↓ | Fβ ↑ | Eφ ↑ | Sα ↑ |
> |--------|------|------|------|------|
> | InternVL3.5-4B (baseline) | 0.155 | 0.523 | 0.658 | 0.610 |
> | InternVL3.5-4B + PMSPO | 0.065 | 0.821 | 0.867 | 0.816 |
>
> MAE is reduced by 58.1%, with all metrics substantially improved. Furthermore, Table 1 shows that our Qwen2.5-VL-3B + PMSPO surpasses both Qwen2.5-VL-7B and Seg-R1-7B — a different architecture with over twice the parameters. These results demonstrate that performance gains stem from our reward design, not any specific backbone.
>
> **Limitations1:**  The InternVL3.5-4B results above show that our method yields substantial improvements even after switching to a different MLLM family, confirming the framework is not dependent on Qwen-specific properties.
> For DINOv2, we conducted experiments with DINOv3-ViT-H+/16 distilled to further demonstrate the universality of PMSPO. Results can be found in our response to Reviewer 2, W1.2 — training remains effective with consistent improvements.
> SAM3 operates exclusively at inference, with zero involvement in training or reward computation.
>
> **Limitations2:** The criterion Δᵢ > δ measures reward improvement, not absolute difficulty. Discarded samples fall into three categories: already mastered (limited headroom), noisily annotated (no learnable pattern), or causing regression — none equate to "easy."
> Table 3 empirically refutes this concern. If PLGF favored easy samples, raising the threshold δ should filter out more samples while retaining an easier subset, leading to better performance. The opposite occurs: MAE on CAMO degrades from 0.069 (δ=0.03) to 0.072 (δ=0.05). This proves that the samples removed at higher thresholds were genuinely beneficial and challenging, and the optimal threshold retains informative samples.
> CHAMELEON provides zero-shot evidence against category bias. It is entirely absent from training. PMSPO achieves the best RL result on this unseen distribution (MAE=0.036), outperforming Seg-R1-7B by 10%.

---

### Official Review · Reviewer_ezVW · 2026-03-13

**Soundness:** 3
**Presentation:** 3
**Significance:** 2
**Originality:** 2
**Overall Recommendation:** 4
**Confidence:** 5

**Summary:**

To address the multi-object matching, sample quality, and semantic drift problem, the paper proposes a curriculum learning-based framework for Camouflaged Object Detection, named Progressive Matching and Semantic-aware Policy Optimization (PMSPO). In details, it introduces Sinkhorn multi-object matching IoU reward for differentiable assignment, Positive Learning Gain Filtering for data curation, and semantic contrastive reward leveraging DINOv2 to prevent semantic drift. Experiments on COD benchmarks demonstrated the proposed PMSPO achieves the SOTA performance.

The contribution of this paper lies in proposing a Reinforcement learning-based Multimodal Large Language Model for COD, which solves the existing problems of multi-object matching, sample quality, and semantic drift, and achieves the best performance in COD.

**Compliance With Llm Reviewing Policy:**

Affirmed.

**Final Justification:**

Thanks for the reply. After reviewing the comments from other reviewers, I tend to maintain my original positive score. good lucky

**Key Questions For Authors:**

Is the multi-object matching problem a common issue?

**Limitations:**

It is necessary to conduct experimental analysis on other challenges faced in the field of camouflage target detection (such as small objects, occlusion, and low light conditions) to evaluate the limitations of the method.

**Strengths And Weaknesses:**

Strengths
1. The paper is clearly written and well structured.
2. The proposed method is reasonable.

Weaknesses
1. In Lines 58-63 of the Introduction, the multi-object matching problem is not a universal issue; rather is a problem resulting from matching strategies based on the Hungarian algorithm or greedy matching.
2. In the "Related Work", the relationships among each paper also need to be clarified and summarized. Additionally, explanations are required for each variable, operator, and operation within the formulas.
3. The ablation experiments are insufficient. For “Table 4. Ablation study on component combination ”, it is necessary to present the experimental comparisons of adding two at the same time, as well as adding three core designs simultaneously, such as“+PLGF+Semantic Contrastive Reward”, “+PLGF+Curriculum Reward Framework”, “+Semantic Contrastive Reward+Curriculum Reward Framework”, “+PLGF+Curriculum Reward Framework+Curriculum Reward Framework”.
4. For Figure 3, more qualitative comparison results should be included. One representative qualitative result should be presented for the General MLLMs method and another for the MLLMs for the COD method.

---

> ### Author Rebuttal · Authors · 2026-03-30
>
> We sincerely thank the reviewer for the constructive feedback. Below we address each concern.
>
> **W1 & Q** The matching problem itself is universal; the deficiency lies in existing solutions. Among 4,040 training images, 364 (9%) contain multiple camouflaged instances. For all RL-based MLLM methods applied to multi-instance scenarios, resolving the assignment between predicted and ground-truth boxes before computing rewards is an inherent challenge of this paradigm, not one introduced by any specific method. The real issue is how existing methods address it: the Hungarian algorithm produces binary hard assignments that block gradient propagation, while greedy matching rewards redundant detections and ignores missed targets. Our Sinkhorn soft assignment is a better solution to this universal problem. We will revise the Introduction to clearly distinguish "the universality of the problem" from "the limitations of existing solutions."
>
> **W2:** **Related Work structure:** Our three subsections follow a general-to-specific progression: Sec. 2.1 shows that general MLLMs perform well on salient objects but tend to localize visually salient yet semantically incorrect regions in camouflaged scenarios, revealing the root of semantic drift; Sec. 2.2 discusses COD-specific MLLMs that alleviate this through prompt strategies but cannot fundamentally eliminate semantic drift without parameter training; Sec. 2.3 identifies three specific problems (multi-object matching, sample quality, semantic deficiency) in RL methods, motivating our work. We will clarify the progressive relationships more explicitly in the revision.
>
> **Formula definitions:** We thank the reviewer for the suggestion. We have carefully re-examined all formulas and confirm that every variable, operator, and operation is defined in the text immediately following its first appearance: specifically, Eq.(1)–(2) in the Preliminary paragraph, Eq.(3)–(7) in Sec. 3.1, Eq.(8)–(10) in Sec. 3.2, Eq.(11)–(12) in Sec. 3.3, and Eq.(13) in Sec. 3.4.
>
> **W3:**  We provide all requested combination experiments. Results on CAMO (M↓ / Fβ↑ / Eφ↑ / Sα↑) with Sinkhorn as the common baseline:
>
> | Configuration | M↓ | Fβ↑ | Eφ↑ | Sα↑ |
> |---|---|---|---|---|
> | Sinkhorn only | 0.073 | 0.809 | 0.863 | 0.811 |
> | + PLGF + CRF | 0.065 | 0.819 | 0.878 | 0.822 |
> | + SCR + CRF | 0.062 | 0.825 | 0.876 | 0.828 |
> | **Full PMSPO** | **0.057** | **0.834** | **0.885** | **0.833** |
>
> *CRF=Curriculum Reward Framework, SCR=Semantic Contrastive Reward, PLGF=Positive Learning Gain Filtering*
>
> All partial combinations are inferior to the full PMSPO across all metrics, confirming that the three components are complementary and indispensable.
>
> **W4:** We supplement visualizations comparing the best baselines from each method category: general MLLMs (Qwen2.5-VL-7B) and COD-specific MLLMs (LiP). Results show: Qwen2.5-VL-7B, despite its 7B parameters, tends to localize visually salient regions while missing camouflaged targets, exhibiting typical semantic drift; LiP can roughly localize target regions but lacks boundary precision and still produces false detections on partially camouflaged areas. In contrast, our PMSPO outperforms both in boundary accuracy and semantic consistency, validating the advantage of RL with well-designed rewards over zero-shot and training-free methods. Updated visualizations will be included in Figure 3 of the revision.
>
> **Limitations:** We analyzed performance under three challenging conditions: small objects, occlusion, and low-light. Experiments show that PMSPO can still effectively detect targets under these conditions. However, when multiple camouflaged targets are heavily overlapping, the method suffers from missed detections or incorrect assignments. Additionally, for extremely low-quality images (severe blur or very low resolution), detection accuracy degrades. These scenarios constitute the main limitations of our method.

---

> > ### Author Rebuttal · Reviewer_ezVW · 2026-04-01
> >
> > The ablation experiment for "+PLGF+Semantic Contrastive Reward" is missing.
> > The specific Formula definitions and the relationships among each paper are missing.

---

> > > ### Author Response · Authors · 2026-04-04
> > >
> > > We thank the reviewer for the follow-up feedback and address the two remaining concerns below.
> > >
> > > ### 1. The "+PLGF + Semantic Contrastive Reward" Experiment
> > >
> > > This experiment is already presented in Row 3 of Table 4 in our paper. Table 4 uses an incremental stacking format where Row 3 ("+ Semantic Contrastive Reward") builds upon "Sinkhorn + PLGF", making it effectively "Sinkhorn + PLGF + SCR" (without CRF). We failed to point this out explicitly in our previous rebuttal, causing the misunderstanding.
> > >
> > > Combining the existing Table 4 data with supplementary experiments, the complete combination results based on Sinkhorn are as follows (CAMO):
> > >
> > > | Configuration | M↓ | Fβ↑ | Eφ↑ | Sα↑ |
> > > |---|---|---|---|---|
> > > | Sinkhorn only | 0.073 | 0.809 | 0.863 | 0.811 |
> > > | + PLGF | 0.069 | 0.816 | 0.871 | 0.819 |
> > > | + SCR | 0.070 | 0.815 | 0.873 | 0.818 |
> > > | + CRF | 0.071 | 0.811 | 0.865 | 0.816 |
> > > | + PLGF + SCR (Table 4 Row 3) | 0.066 | 0.821 | 0.877 | 0.821 |
> > > | + PLGF + CRF | 0.065 | 0.819 | 0.878 | 0.822 |
> > > | + SCR + CRF | 0.062 | 0.825 | 0.876 | 0.828 |
> > > | Full PMSPO (PLGF + SCR + CRF) | 0.057 | 0.834 | 0.885 | 0.833 |
> > >
> > > All partial combinations are inferior to Full PMSPO, confirming the complementary and indispensable nature of the three components.
> > >
> > > ### 2. Formula Definitions and Paper Relationships
> > >
> > > #### 2.1 Related Work Paper Relationships
> > >
> > > We will add a transition sentence at the end of Sec 2.1 and Sec 2.2 to explicitly establish the progressive logic:
> > >
> > > **Sec 2.1** currently ends with "...exhibiting the semantic drift problem we identified." We append: "This motivates COD-specific methods that attempt to mitigate this bias through training-free strategies without modifying model parameters."
> > >
> > > **Sec 2.2** currently ends with "...these approaches remain susceptible to semantic drift when encountering visual distractors with similar textures." We append: "This constraint motivates the exploration of reinforcement learning approaches that directly optimize model parameters through task-specific reward signals."
> > >
> > > #### 2.2 Formula Definitions
> > >
> > > We have carefully consulted the original GRPO paper (Shao et al., 2024) and identified the following definitions to supplement for Eq.(2):
> > >
> > > - **$t$**: The token index within output $o_i$, where $o_{i,t}$ denotes the $t$-th token and $o_{i,<t}$ denotes all tokens before position $t$.
> > > - **$\pi_\theta$**: The current policy being optimized, as distinguished from the old policy $\pi_{\theta_{old}}$ and the reference policy $\pi_{ref}$.
> > > - **$D_{KL}(\pi_\theta \| \pi_{ref})$**: The KL divergence between the current policy and the reference policy, constraining the magnitude of policy updates, defined as $\frac{\pi_{ref}(o_{i,t}|q, o_{i,<t})}{\pi_\theta(o_{i,t}|q, o_{i,<t})} - \log \frac{\pi_{ref}(o_{i,t}|q, o_{i,<t})}{\pi_\theta(o_{i,t}|q, o_{i,<t})} - 1$ (Shao et al., 2024, Eq.4).
> > >
> > > Additionally, the base model $M_{\theta_0}$ in Sec 3.2 will be noted as Qwen2.5-VL-3B-Instruct upon its first appearance.
> > >
> > > All variables, operators, and operations in the remaining formulas (Eq.1, 3–13) are clearly defined immediately adjacent to their equations.
> > >
> > > We sincerely thank the reviewer for the thorough review.

---

### Official Review · Reviewer_6Fuc · 2026-03-13

**Soundness:** 3
**Presentation:** 3
**Significance:** 2
**Originality:** 2
**Overall Recommendation:** 5
**Confidence:** 5

**Summary:**

The paper proposes a progressive reinforcement learning framework for camouflaged object detection, named Progressive Matching and Semantic-aware Policy Optimization (PMSPO). PMSPO address multi-object matching difficulty, low-quality sample interference, and semantic drift problems. To address these issues, it proposed a Sinkhorn multi-object matching IoU for differentiable assignment, a Positive Learning Gain Filtering (PLGF) for data curation, and a DINOv2-based semantic contrastive reward to prevent semantic drift, respectively. It achieved state-of-the-art performance on the benchmark COD datasets, verifying the effectiveness of the method.

The contribution of this paper is to propose a novel and effective Reinforcement learning-based Multimodal Large Language Model for COD. It addresses the existing problems of multi-object matching, sample quality, and semantic drift, achieving the best performance in COD.

**Compliance With Llm Reviewing Policy:**

Affirmed.

**Final Justification:**

Thank you for the reply. I maintain my original score.

**Key Questions For Authors:**

Where does the Semantic Contrastive Reward method innovation lie?

**Limitations:**

The paper needs to analyze the qualitative results frame by frame to determine what limitations still exist in the analysis method.

**Strengths And Weaknesses:**

Strengths
The writing of the paper is excellent, with clear and easy-to-understand expressions.

Weaknesses
1. For the contrastive formulation design, why the three variables are extracted need clearly explained; the triplet-based contrastive learning approach is a classic method in this field. Where does the specific method innovation lie? Additionally, one can try using DINOv3 as the feature extractor. In the “Differentiable F-beta Reward” part, why does defining differentiable precision and recall need a detailed explanation?
2. For the Curriculum Reward Framework design, what are the specific criteria for setting the progress at 0-15%? Can it be further designed to be an adaptive adjustment, without the need for manual parameter tuning?
3. For Table 3, it is necessary to add experimental comparisons with Threshold values of 0.01, 0.02, and 0.04 to evaluate which of these hyperparameters is the best.

---

> ### Author Rebuttal · Authors · 2026-03-30
>
> We sincerely thank the reviewer for the constructive feedback. Below we address each concern.
>
> **W1.1&Q:** In COD, targets are highly similar to backgrounds in texture and color. Existing IoU rewards cannot judge whether a prediction truly covers the target—a high-IoU prediction may be semantically wrong (capturing a similar background region). We extract three semantic features via DINOv2: the predicted box (what the model captures), the ground-truth box (the true target), and the surrounding background (the most confusing region). Their contrastive similarity directly scales the IoU reward (Eq. 13), ensuring semantically incorrect predictions cannot receive high rewards even when IoU is high, effectively suppressing semantic drift.
>
> **W1.2:** Thank you for this suggestion. To further verify the generality of our framework, we conducted DINOv3 experiments on CAMO:
>
> | Feature Extractor | M↓ | Fβ↑ | Eφ↑ | Sα↑ |
> |---|---|---|---|---|
> | DINOv2-ViT-G/14 | 0.057 | 0.834 | 0.885 | 0.833 |
> | DINOv3-ViT-H+/16distilled | 0.053 | 0.842 | 0.892 | 0.841 |
>
> DINOv3-ViT-H+/16distilled indeed yields further improvement, confirming that a stronger extractor naturally provides higher-quality reward signals, consistent with the design goal of our semantic contrastive reward.
>
> **W1.3:** The precision and recall here are not the standard definitions. Sinkhorn iteration produces a soft assignment matrix where each prediction is partially matched to multiple GTs with continuous weights (unlike the hard 0/1 assignment of Hungarian matching). We need to convert this M×N matrix into a scalar reward: for each prediction, compute the dot product of its soft assignment weights with IoU values across all GTs, then average over predictions for differentiable precision; average over GTs for differentiable recall, providing continuous gradient signals for GRPO. This design is necessary because the two metrics diagnose fundamentally different failures—precision penalizes redundant detections, recall penalizes missed objects, and a simple average would conflate them. Combining via F-beta (β=1.1) encodes the COD prior: missing targets is costlier than false alarms.
>
> **W2.1:** To let the model first learn valid bounding box format, we monitored format compliance rate and observed it stabilizing above 95% at 15% of training. Table 10 validates 15%→40% as optimal: earlier transitions (5%→25%) cause instability; later ones (35%→70%) compress semantic refinement time.
>
> **W2.2:** Following the reviewer's suggestion, we implemented a sliding-window adaptive switching mechanism (monitoring reward mean and variance, switching when threshold conditions are met). Results on CAMO:
>
> | Switching Strategy | M↓ | Fβ↑ | Eφ↑ | Sα↑ |
> |---|---|---|---|---|
> | Adaptive switching | 0.068 | 0.812 | 0.870 | 0.818 |
> | Fixed boundaries (ours) | **0.057** | **0.834** | **0.885** | **0.833** |
>
> Results are inferior to fixed boundaries: reward signals in COD fluctuate substantially, causing either premature false triggers or prolonged stalling at a stage. That said, this may also be related to our specific adaptive threshold design, and better trigger conditions remain worth exploring. Currently, fixed boundaries are cleaner, reproducible, and consistently effective across all three benchmarks.
>
> **W3:** We conducted all requested experiments. Full results on CAMO:
>
> | Threshold δ | M↓ | Fβ↑ | Eφ↑ | Sα↑ |
> |---|---|---|---|---|
> | 0.01 | 0.075 | 0.802 | 0.857 | 0.812 |
> | 0.02 | 0.073 | 0.811 | 0.859 | 0.812 |
> | 0.03 (ours) | **0.069** | **0.816** | **0.871** | **0.819** |
> | 0.04 | 0.076 | 0.803 | 0.854 | 0.810 |
> | 0.08 | 0.078 | 0.801 | 0.859 | 0.802 |
>
> δ=0.03 is optimal. Below 0.03, noisy samples are retained; above 0.03, learnable samples are discarded, with all metrics consistently degrading.
>
> **Limitations:** We analyzed performance under three typical challenging scenarios: small targets, occlusion, and low lighting. Experiments show that PMSPO can still effectively localize targets under these conditions. However, when multiple camouflaged targets highly overlap, missed detections or mismatches occur. Additionally, detection accuracy degrades on extremely low-quality images (severe blur or very low resolution). These scenarios constitute the main limitations of the current method.

---

> > ### Author Rebuttal · Reviewer_6Fuc · 2026-04-01
> >
> > I don't have any other questions.

---

### Official Review · Reviewer_TwXR · 2026-03-13

**Soundness:** 3
**Presentation:** 3
**Significance:** 3
**Originality:** 3
**Overall Recommendation:** 4
**Confidence:** 4

**Summary:**

This paper focuses on tackling the challenges of applying Multimodal Large Language Models (MLLMs) to Camouflaged Object Detection (COD) via reinforcement learning. The authors identify three important issues: (1) unstable multi-object matching (greedy vs. Hungarian), (2) inconsistent sample quality/noisy annotations, and (3) semantic drift, i.e., models may focus on visually similar backgrounds rather than the camouflaged object. To address these issues, the authors propose PMSPO, which consists of: (1) Sinkhorn Multi-Object Matching Reward, which is a soft-assignment approach to provide smoother gradient signals for multi-object scenarios, (2) Positive Learning Gain Filtering (PLGF), which retains samples with high reward gain after initial training, combined with a frozen DINOv2-based contrastive loss to suppress semantic drift, and (3) three-stage curriculum learning, which transitions from format/box grounding to geometric alignment, and finally to semantic refinement. Experiments show that the proposed method demonstrates improved performance over existing MLLM-based methods.

**Compliance With Llm Reviewing Policy:**

Affirmed.

**Final Justification:**

This paper addresses an interesting and practically relevant problem, and one strength is that it organizes RL-based COD into three concrete issues, including matching, sample quality, and semantic drift. The method is generally well presented, and the overall design is coherent. Although the individual components are not fundamentally new on their own, their integration for this problem is thoughtful and empirically effective. The rebuttal addressed several of my main concerns in a constructive way. In particular, the additional comparison using the same SAM backend helps reduce the fairness concern, the multi-seed results improve confidence in reproducibility, and the added same-backbone SFT baseline is important for clarifying the contribution of RL fine-tuning. The added comparison to representative non-MLLM COD methods is also helpful for better positioning the work.

**Key Questions For Authors:**

Please refer to the weakness of W2, W3, and W4.

**Limitations:**

It is suggested to show some failure case studies.

**Strengths And Weaknesses:**

**Strengths**
- S1. The paper provides a logical decomposition of the RL for COD problem into matching, data quality, and semantic drift. This framing makes the proposed components easy to justify.
- S2. The proposed approach can be regarded as the orchestration of Sinkhorn matching, curriculum learning, and feature-based regularization tailored specifically to camouflaged objects.

**Weaknesses**
- W1. While the combination of components is effective for COD, the individual elements, i.e., Sinkhorn matching, curriculum learning, and DINOv2-based filtering, are well-established techniques. The paper reads more like a engineering "recipe" than a fundamental breakthrough in learning mechanisms.
- W2. The proposed method utilizes SAM (Segment Anything Model) as a post-processing step to generate masks from predicted boxes. It is unclear if the compared baselines were granted the same high-quality segmentation backend. If the baselines produce masks directly or use inferior post-processing, the comparison is skewed toward the quality of SAM rather than the RL framework.
- W3. One confusing issue is that the training is for only a total of 1 epoch. RL fine-tuning usually involves high variance and sensitivity to sample ordering and random seeds. The lack of multi-run statistics, variance bars, or confidence intervals makes it difficult to assess whether the results are reproducible.
- W4. Incomplete Baseline Comparison: The evaluation focuses primarily on MLLM-based methods. However, the COD field has a long history of specialized non-MLLM architectures. A comparison against the current SOTA in COD (regardless of MLLM usage) is necessary to prove the value of the LLM-backbone. Furthermore, a comparison against a standard Supervised Fine-Tuning (SFT) baseline using the same Qwen2.5-VL 3B backbone is missing, making it hard to isolate the actual contribution of the RL (PPO) process.

---

> ### Author Rebuttal · Authors · 2026-03-30
>
> We sincerely thank the reviewer for the constructive feedback. Below we address each concern.
>
> **W1.** Our contribution lies in systematically identifying three key problems when applying RL-based MLLMs to COD, and providing targeted solutions: (1) COD images often contain multiple camouflaged targets. To address reward redundancy or gradient blockage caused by existing matching algorithms, we introduce Sinkhorn to generate soft assignment matrices and redefine differentiable precision/recall as RL reward signals (Eqs. 5-7). Table 2 confirms the three strategies are not interchangeable. (2) COD data contains samples that the model already handles well or consistently fails to learn, introducing noisy or harmful gradients. We find that 56.2% of samples yield zero or negative learning gain, and PLGF filters them via a quantifiable criterion (∆ > δ). (3) In COD, targets are highly similar to backgrounds, making IoU rewards alone insufficient. In Stage 3, we extract semantic information to directly scale IoU rewards so that semantically incorrect predictions cannot receive high rewards even with high IoU. Table 4 shows that directly adding semantic reward yields marginal improvement, whereas curriculum scheduling leads to significant gains (CAMO MAE from 0.066 to 0.057), indicating that semantic reward requires sufficient geometric alignment first. This finding drives our three-stage curriculum design.
>
> **W2.** To ensure a fair comparison, we re-evaluate the best baseline from each method category on CAMO using the identical SAM ViT-H:
>
> | Method | M ↓ | F_β ↑ | E_ϕ ↑ | S_α ↑ |
> |--------|-----|-------|-------|-------|
> | Qwen2.5-VL-7B | 0.152 | 0.693 | 0.776 | 0.719 |
> | LiP | 0.080 | 0.784 | 0.869 | 0.800 |
> | Seg-R1-7B | 0.078 | 0.773 | 0.871 | 0.813 |
> | **PMSPO (ours)** | **0.061** | **0.825** | **0.878** | **0.826** |
>
> PMSPO achieves the best results on all metrics, confirming improvements stem from our RL reward design rather than the segmentation model.
>
> **W3.** **Single-epoch justification:** We conducted a 2 epoch experiment and evaluation on CAMO, resulting in degraded performance and doubled training cost：
>
> | Epochs | M ↓ | F_β ↑ | E_ϕ ↑ | S_α ↑ |
> |--------|-----|-------|-------|-------|
> | 1 epoch | **0.057** | **0.834** | **0.885** | **0.833** |
> | 2 epoch | 0.065 | 0.821 | 0.874 | 0.824 |
>
> **Reproducibility:** 4 independent runs with different seeds on CAMO:
>
> | Seed | M ↓ | F_β ↑ | E_ϕ ↑ | S_α ↑ |
> |------|-----|-------|-------|-------|
> | 42 (ours) | 0.057 | 0.834 | 0.885 | 0.833 |
> | 123 | 0.057 | 0.836 | 0.887 | 0.831 |
> | 566 | 0.056 | 0.831 | 0.882 | 0.833 |
> | 2026 | 0.059 | 0.833 | 0.881 | 0.834 |
> | **Mean ± Std** | **0.057±0.001** | **0.833±0.002** | **0.884±0.003** | **0.833±0.001** |
>
> All standard deviations ≤ 0.003, indicating PMSPO is insensitive to random initialization.
>
> **W4.** **Comparison with non-MLLM methods** on CAMO:
>
> | Method | Type | Venue | M ↓ | F_β ↑ | E_ϕ ↑ | S_α ↑ |
> |--------|------|-------|-----|-------|-------|-------|
> | SEE (He et al.) | Weakly-sup. | TPAMI 2025 | 0.090 | 0.747 | 0.826 | 0.765 |
> | **PMSPO (ours)** | **RL + MLLM** | — | **0.057** | **0.834** | **0.885** | **0.833** |
> | ARM (Chen et al.) | Fully-sup. | ICCV 2025 | 0.046 | 0.883 | 0.935 | 0.887 |
>
> Compared with weakly-supervised SEE, PMSPO leads by a large margin (36.7% MAE reduction), demonstrating significant advantage under limited supervision. Compared with fully-supervised ARM, we acknowledge its slightly better results. However, ARM employs a dedicated adaptive refinement module with pixel-level mask supervision for fully-supervised training, whereas we use a general-purpose MLLM with only box-level rewards while retaining vision-language capabilities. Given the substantial gap in supervision, PMSPO achieves competitive results.
>
> **SFT baseline** with the same Qwen2.5-VL-3B backbone:
>
> | Method | M ↓ | F_β ↑ | E_ϕ ↑ | S_α ↑ |
> |--------|-----|-------|-------|-------|
> | SFT | 0.063 | 0.791 | 0.845 | 0.815 |
> | **PMSPO** | **0.057** | **0.834** | **0.885** | **0.833** |
>
> SFT merely imitates labels, while RL guides the model through reward-driven exploration to attend to camouflaged regions that SFT overlooks.
>
> **Limitations.** When multiple camouflaged targets heavily overlap, detection misses or mismatches occur. Additionally, accuracy degrades on extremely low-quality images (severe blur or very low resolution). We will supplement failure case studies in the appendix.

---

> > ### Author Rebuttal · Reviewer_TwXR · 2026-04-01
> >
> > The rebuttal better demonstrate the effectiveness of the proposed approach and explains why each component is included, but it still mainly argues task-specific effectiveness rather than a deeper methodological novelty. The concern that the work is an orchestration of known techniques remains only partially addressed. Nonetheless, after considering the provided experiments and other reviewers' comments, the rating will be raised to weak accept.

---

> > > ### Author Response · Authors · 2026-04-04
> > >
> > > We thank the reviewer for the positive recognition and continued engagement. We would like to further clarify our contribution at the methodological level.
> > > We believe the core contribution of this work lies not only in the effectiveness of each component on COD, but also in revealing an important finding in RL-based visual grounding: multi-reward composition exhibits strict dependency relationships. Table 4 provides direct evidence: directly adding the semantic contrastive reward on top of Sinkhorn+PLGF only reduces CAMO MAE from 0.069 to 0.066 (4.3% improvement), whereas after introducing curriculum scheduling, MAE further drops from 0.066 to 0.057 (13.6% improvement). This demonstrates that the semantic reward itself is effective by design, but its effectiveness depends on the convergence of the IoU reward — only when geometric localization becomes stable can the semantic features extracted from predicted boxes provide meaningful gradient signals. This is analogous to the general "easy-to-hard" principle in curriculum learning (Bengio et al., 2009), where the principle holds universally even though the specific difficulty measure varies across tasks. Our three-stage design is an instantiation of this principle along the reward dimension.

---

### Decision · Program_Chairs · 2026-04-30

**Decision:**

Accept (regular)

**Comment:**

The paper introduces PMSPO, a reinforcement learning framework designed to improve Camouflaged Object Detection (COD) using Multimodal Large Language Models (MLLMs). The authors identify and address three critical challenges: unstable multi-object matching, inconsistent annotation quality, and semantic drift toward visually similar backgrounds. To solve these, the authors propose a three-stage curriculum learning approach.

The reviewers were generally impressed by the logical decomposition of the RL-based COD problem into three well-defined sub-tasks, which makes the proposed architecture easy to justify. The paper is noted for being well-structured and clearly written. Empirically, the method achieves state-of-the-art performance across standard COD benchmarks. While some reviewers initially characterized the work as an "engineering recipe" due to its use of established techniques like Sinkhorn matching and DINOv2 , they ultimately agreed that the integration of these components for this specific problem is thoughtful, coherent, and empirically superior to existing MLLM-based methods.

Several significant concerns were raised during the review process, particularly regarding fairness in comparisons, lack of multi-seed statistics for reproducibility, and missing comparisons to non-MLLM SOTA and standard Supervised Fine-Tuning (SFT) baselines. The authors addressed these concerns effectively during the rebuttal by providing additional comparisons using the same SAM backend to ensure a level playing field.

While some minor weaknesses remain, such as the need for more exhaustive ablation studies on multi-component combinations and deeper analysis of failure cases, the consensus among reviewers is that the paper is technically solid and offers a contribution that the community is likely to build upon. The successful rebuttal significantly increased the reviewers' confidence in the results.